# Integrated Correction Algorithm for X Band Dual-Polarization Radar Reflectivity Based on CINRAD/SA Radar

**Chao Wang [1,2], Chong Wu [2,\*], Liping Liu [2], Xi Liu [3,4] and Chao Chen [5]**

[1] School of Atmospheric Physics, Nanjing University of Information Science and Technology, Nanjing 210044, China; chaowang716@163.com

[2] Key Laboratory of Severe Weather, Chinese Academy of Meteorological Sciences, Beijing 100081, China; liulp@cma.gov.cn

[3] Key Laboratory of Transportation Meteorology, China Meteorological Administration, Nanjing 210009, China; xiliu@nuist.edu.cn

[4] Jiangsu Institute of Meteorological Sciences, Nanjing 210009, China

[5] Guangdong Meteorological Observatory, Guangzhou 510640, China; j1alin@163.com

\* Correspondence: wuchong@cma.gov.cn; Tel.: +86-10-5899-4251

**Abstract:** The values of ratio *a* of the linear relationship between specific attenuation and specific differential phase vary significantly in convective storms as a result of resonance scattering. The best-linear-fit ratio *a* at X band is determined using the modified attenuation correction algorithm based on differential phase and attenuation, as well as the premise that reflectivity is unattenuated in S band radar detection. Meanwhile, the systemic reflectivity bias between the X band radar and S band radar and water layer attenuation ($Z_W$) on the wet antenna cover of the X band radar are also considered. The good performance of the modified correction algorithm is demonstrated in a moderate rainfall event. The data were collected by four X band dual-polarization (X-POL) radar sites, namely, BJXCP, BJXFS, BJXSY, and BJXTZ, and a China's New Generation Weather Radar (CINRAD/SA radar) site, BJSDX, in Beijing on 20 July 2016. Ratio *a* is calculated for each volume scan of the X band radar, with a mean value of 0.26 dB deg$^{-1}$ varying from 0.20 to 0.31 dB deg$^{-1}$. The average values of systemic reflectivity bias between the X band radar (at BJXCP, BJXFS, BJXSY, and BJXTZ) and S band radar (at BJSDX) are 0, −3, 2, and 0 dB, respectively. The experimentally determined $Z_W$ is in substantial agreement with the theoretically calculated ones, and their values are an order of magnitude smaller than rain attenuation. The comparison of the modified attenuation correction algorithm and the empirical-fixed-ratio correction algorithm is further evaluated at the X-POL radar. It is shown that the modified attenuation correction algorithm in the present paper provides higher correction accuracy for rain attenuation than the empirical-fixed-ratio correction algorithm.

**Keywords:** X band dual-polarization radar; rain attenuation correction; water layer attenuation

## 1. Introduction

A concept of a radar network refers to using several small-antenna radars to cover an area of interest, and the radar network with high measurement sensitivity and high spatial and temporal resolution is a new approach for observing precipitation [1,2]. A short-range radar network consisting of several X band dual-polarization (X-POL) radars in Beijing could effectively improve radar usage in urban flood monitoring, short-term nowcast, and weather modification. However, it is well known that convective storms cause significant attenuation (at horizontally polarized waves) and differential attenuation (between horizontal and vertical polarized waves) at X band (wavelength of

3 cm). The specific attenuation ($A_H$) of the X band radar at a certain rain rate is about 7–10 times larger than that of C and S band radars (wavelengths of 5 and 10 cm) [3,4]. Additional attenuation could also be caused by the water layer on the protective cover of the radar antenna [5,6]. As a consequence, measurements of the reflectivity factor ($Z$) and differential reflectivity factor ($Z_{DR}$) of X band radars must be corrected for rain attenuation before they can be used quantitatively in rainfall estimation algorithms, hydrometeor identification, or numerical assimilation [3,4].

Simplified versions of the correction method based on specific differential phase $K_{DP}$ (or differential phased $\Phi_{DP}$) was introduced by Bringi et al., 1990 [3] due to the fact that $K_{DP}$ and the $\Phi_{DP}$ are immune to attenuation. Bringi et al., 1990 [3] also found the radar frequencies under 10 GHz (e.g., S, C, and X band radars) that reflect nearly linear relations between $A_H$ and $K_{DP}$, where $A_H$ is specific attenuation of microwave radiation at horizontal polarization. In this technique, the determination of ratio $a = A_H/K_{DP}$ is sophisticated and generally assumes a drop in the aspect ratio (i.e., minor-to-major dimension ratio) vs. size relations and temperature by scattering simulations of disdrometer data [3,7–10]. However, the disdrometer data may only represent a small observation area of raindrop size distributions (RSDs) near the ground, and a bias between the simulated ratio $a$ and the desired one always exists and results in inaccuracy correction of rain attenuation.

In rain, ratios $a$ at X band usually vary from 0.17 to 0.31 dB deg$^{-1}$. High variabilities especially occur in convective storms containing large raindrops and hail owing to the effects of resonance scattering, as reported by Bringi et al., 1990, Jameson. 1992, Matrosov et al., 2002, Park et al., 2005, and Testud et al., 2000 [3,7–11]. In general, ratio $a$ is in positive correlation with raindrop size and is negative correlation with rain temperature [3,7,12]. Gu et al., 2011 [12] also implied that the ratio $a$ strongly depends on temperature, but it is much less sensitive to differential reflectivity $Z_{DR}$, which is related to the raindrop diameter.

Some more sophisticated attenuation correction approaches take into account the variability of ratio $a$ as well [9,11,13–16]. Bringi et al., 2001 [13] proposed a self-consistent (SC) method, which iteratively checks $\Phi_{DP}$ in each beam resolution volume along each ray and constrains the corrected $Z_{DR}$ on the far side of the rain cell in order to obtain the optimal ratio $a$ at each particular ray. Park et al., 2005 [9] further extended this SC technique to X band measurements. However, a radar may not be able to observe the far side of the rain cell if the rainfall regions exceed the radar maximum observation range. Moreover, Vulpiani et al., 2008 [16] further completed the variabilities of ratio $a$ in each beam resolution volume along the ray. However, this attenuation correction at any ray only uses a fixed correction factor weighted by $K_{DP}$. In addition, the correction approach based on $\Phi_{DP}$ suggested by Bringi et al., 1990 [3], used especially in a rain area consisting of large-size raindrop particles in convective storms, was modified by Carey et al., 2000 [14]. This modified approach derives variable ratios $a$ by using least squares linear regression the identified big drop consisting of polarimetric measurements. The concept of Carey et al., 2000 [14] was further advanced by Ryzhkov et al., 2007 [15]. All in all, reliable regression needs to consider many thresholds in big drop identification algorithms.

S band reflectivity attenuation in rain is typically more than one order of magnitude smaller than it is in X band attenuation and can be usually neglected compared with X band attenuation in dual-frequency measurements [17]. Matrosov et al., 2014 [11] modified the attenuation correction technique suggested by Bringi et al., 1990 [3] to the dual-wavelength observations collected by the CSU-CHILL radar (detailed radar technique parameters introduced in [11]) whose X and S band channels share the same antenna system. The advantages of the modified technique are essentially free of modeling assumptions about their drop shapes, oscillation modes, size distribution, and relations between attenuation characteristics and reflectivity.

Since 2015, the Beijing Meteorological Service has built multiple X-POL radars (wavelength λ ~ 3 cm) in the observation coverage area of CINRAD/SA radar (λ ~ 10 cm) in Daxing district. However, rain attenuation at X band severely affects the usage of reflectivity. In this study, on the basis of the attenuation correction method suggested by Bringi et al., 1990 [3] and the premise that the reflectivity is unattenuated in S band radar detection, a modified approach improves the

quality of X band reflectivity, including rain attenuation correction, additional attenuation correction for water layer on the protective cover of radar antenna, and systemic reflectivity bias between the X and S band radars.

This work is organized into five sections. Section 2 mainly describes the reflectivity integrated correction (RIC) method in detail, and it also briefly introduces the performance parameters of the meteorological radar at X and S bands and the time periods of precipitation observation. First, Section 3, lists the values of ratio *a* at X band from scattering simulations and experiments, and these listed ratios are used to validate the ratios obtained by the modified approach from this article. Second, the relation between the thickness of the water layer and attenuation is used to calculate theoretical water layer attenuation with a different rainfall rate, and the results are compared with the experimental ones obtained in this article to verify the truth. Third, this Section also analyzes the disparities in the systemic reflectivity bias between four X-POL radars and the same CINRAD/SA radar. In Section 4, the correction of rain attenuation is evaluated. Section 5 summarizes results and presents the conclusions.

## 2. Observations and RIC Algorithm

### 2.1. Observations

The rainfall data used in this study was collected by four X-POL radars with the same technology parameters and China's New Generation Weather Radar (CINRAD/SA radar) operated by Beijing Metstar Radar Co., Ltd. (Beijing, China). The project of observing rainfall in Beijing, China, was carried out in July and August 2016. The main specifications of CINRAD/SA and X-POL radars are listed in Table 1. The X-POL radar stations are located in Changping, Fangshan, Shunyi, and Tongzhou districts of Beijing (BJXCP, BJXFS, BJXSY, and BJXTZ sites, respectively), and the CINRAD/SA radar is located in Daxing district of Beijing (BJSDX site), just in the center of four X-POL radar sites. The CINRAD/SA radar underwent online automatic calibration after each volume scan [18]. The positions of five radars and terrain are shown in Figure 1.

**Table 1.** System characteristics of CINRAD/SA radar and X-POL radar.

| Parameter | X-POL Radar | CINRAD/SA Radar |
|:---:|:---:|:---:|
| Frequency | 9300–9500 MHz | 2700–3000 MHz |
| Antenna cover diameter | $\geq$4 m | 11.9 m |
| Polarization | Linear H and V | Linear H |
| Volume coverage patterns | VCP 21 | VCP 21 |
| Time of VCP 21 | 4 min | 6 min |
| Range resolution | 75 m | $Z$ (1000 m), $V_D$ and $W$ (250 m) |
| Observation range | 90 km | $Z$ (230 km), $V_D$ and $W$ (150 km) |
| Measurement accuracy | $Z$ ($\leq$ 1), $V_D$ ($\leq$ 1), $W$ ($\leq$ 1), $\rho_{HV}(0)$ ($\leq$ 0.01), $Z_{DR}$ ($\leq$ 0.2), $\Phi_{DP}$ ($\leq$ 3), $K_{DP}$ ($\leq$ 0.2) | $Z$ ($\leq$ 1), $V_D$ and $W$ ($\leq$ 1) |

Reflectivity factor ($Z$; dBZ), doppler velocity ($V_D$; m/s), spectral width ($W$; m/s), cross-correlation correlation coefficient ($\rho_{HV}(0)$), differential reflectivity factor ($Z_{DR}$; dB), differential phase ($\Phi_{DP}$; deg), and specific differential phase ($K_{DP}$; deg/km).

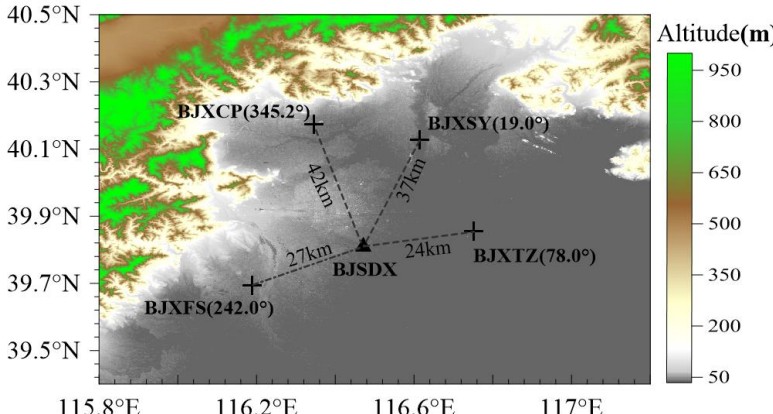

**Figure 1.** Distribution of radars (plus signs and triangles) and topography (shadings) over Beijing and its vicinity. Four plus signs indicate the locations of four X-POL radar sites (BJXFS, BJXCP, BJXSY, and BJXTZ). The triangle shows the location of BJSDX site. Distance and azimuth (0 deg in the north) of each X-POL radar relative to the BJSDX site are labeled.

## 2.2. RIC Algorithm

Figure 2 shows the theoretical diagram of RIC algorithm. The site locations (height, latitude, and longitude) of X-POL and CINRAD/SA radars are used to find a proximate pixel of CINRAD/SA radar for each pixel of X-POL radar. Forming each pair of pixels therefore includes some observed information, such as polarimetric measurements of X-POL and CINRAD/SA radars. The S band reflectivity values are notably unattenuated in contrast to X band ones, within shorter ranges [17]. Based on the linear $PIA$-$\Phi_{DP}$ relation in Equation (4), utilizing the long-term collected pixel pairs (including S band reflectivity, X band reflectivity, and differential phase) can be expressed as Equation (4) by using least squares linear regression to obtain an optimal pair of ratio $a$ and intercept $\Delta Z_0$.

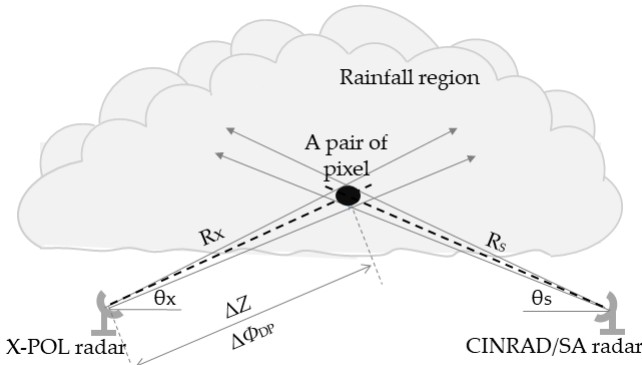

**Figure 2.** Theoretical diagram of RIC methods. The pair of pixels (black ellipse) in the rainfall region respectively comes from the X-POL radar and the CINRAD/SA radar. Since attenuation generally increases with distance, to ensure that the observation time between two radars is close and the distance and height of X band and S band pixels is equal, the basic conditions to form a pair of pixels are as follows: (1) the absolute difference between the distance ($R_X$) from the X-POL radar pixel to the X-POL radar station and the distance ($R_S$) from the CINRAD/SA radar pixel to the CINRAD/SA radar station is less than 10 km, i.e., $|R_X - R_S| < 10$ km; (2) the absolute difference of volume scanned time between X-POL and CINRAD/SA radars are within 2 min, i.e., $|T_X - T_S| < 2$ min; (3) the absolute difference of the viewing angle between CINRAD/SA and X-POL radars is less than 0.01, i.e., $|\theta_X - \theta_S| < 0.01$ deg. $\Delta Z$ is the total rain attenuation value of X band reflectivity on the path of long $R_X$, i.e., path-integrated attenuation $PIA$ in Equation (2), and $\Delta Z$ can here be approximately replaced with the difference between S band reflectivity and X band reflectivity in the pixel pair, i.e., $\Delta Z(S–X)$ in Equation (4). $\Delta \Phi_{DP}$ is the corresponding change of $\Phi_{DP}$ on the path of long $R_X$.

The X band relationship between specific attenuation $A_H$ and $K_{DP}$ is expressed as [3,8,13]:

$$A_H = a \times K_{DP},\tag{1}$$

where ratio $a$ is the rain–attenuation correction coefficient and its unit is dB deg$^{-1}$. Based on Equation (1), the path-integrated attenuation *PIA* is calculated by:

$$PIA = a \times \Delta\Phi_{DP},\tag{2}$$

and

$$\Delta\Phi_{DP} = \Phi_{DP}(X) - \Phi_{DP}(0).\tag{3}$$

In Equation (3), $\Phi_{DP}(X) - \Phi_{DP}(0)$ is a continuously increased X band differential phase value with increasing range and $\Phi_{DP}(0)$ (unit: deg) is the estimated initial system phase at each radial due to radar hardware [11,14]. The reflectivity difference $\Delta Z(S–X)$ between S and X band pixels in the same space-time, with approximately equal horizontal distances and heights, is approximately equal to the X band *PIA* at horizontal polarization. Therefore, considering the systemic reflectivity bias between two radars and the attenuation of wet antenna cover $\Delta Z_0$, Equation (2) is expressed as:

$$\Delta Z(S - X) = a \times \Delta\Phi_{DP}(X) + \Delta Z_0.\tag{4}$$

$\Delta Z_0$ is given as:

$$\Delta Z_0 = \begin{cases} Z_D + Z_W, & \text{rainfall regions stay on the radar site} \\ Z_D, & \text{rainfall regions do not stay on the radar site} \end{cases},\tag{5}$$

where $Z_D$ is in dB and can be termed the systemic reflectivity bias between CINRAD/SA and X-POL radars, and $Z_W$ (unit: dB) is another attenuation except the rain caused by the water layer on the X-POL radar antenna cover. The RIC methods for reflectivity correction are as follows:

(a)　Preprocessing for S band and X band radar data:

The median filter technique used in observed $\Phi_{DP}$ to obtain filtered $\Phi_{DP}$ and the value of $\Phi_{DP}(0)$ at each radial is calculated by Xiao's method [19]. The non-meteorological echoes, such as ground clutters and insect or bird clutters, are distinguished by using the cross-correlation correlation coefficient $\rho_{HV}(0)$ [20].

(b)　Collections of pixel pairs between X and S bands:

To ensure the temporal and spatial matches between the X-POL and CINRAD/SA radar pixels, data with less than 2 min time differences of volume scans between X-POL and CINRAD/SA radars are selected. Meanwhile, the pixel pairs including an X-POL radar pixel and a CINRAD/SA radar pixel in the rainfall region are collected, in which the difference between the observed slope length of X-POL radar pixel ($R_X$) and the one of CINRAD/SA radar pixel ($R_S$) is less than 10 km under the bright band, whereas the viewing angle difference between two radars is not more than 0.01 deg [9,21]. Therefore, the collected pixels of CINRAD/SA and X-POL in each pair are in the same height and distance as much as possible (i.e., $R_X \approx R_S$, $R_X \sin(\theta_X) \approx R_S \sin(\theta_S)$ and see Figure 2) to reduce the adverse impact due to the spatial position difference.

(c)　Calculations of ratio a and intercept $\Delta Z_0$:

These pixel pairs based on radar measurements are used to determine a pair of optimal coefficients (i.e., ratio $a$ and intercept $\Delta Z_0$) by least squares linear regressions. In this study, RIC methods are subdivided into dynamic RIC and static RIC. The dynamic RIC can continuously obtain a pair of coefficients for all current observation data in real time. On the other hand, static RIC can obtain a pair of averaged coefficients using observed data over previous time periods, such as an hour or

more, respectively termed as static RIC (an hour) and static RIC (a rain event). In linear regression, the determination of an optimal DP relation needs to consider (1) the difference in range resolution between two kinds of radars, (2) the magnitude values of $\Phi_{DP}$ at X band, and (3) non-meteorological clutters. Therefore, the following regression conditions are summarized by comparing fitting results under different conditions, namely, the correlation coefficient $CC_{NH}$ in Equation (6) and root-mean-square error *RMSE* in Equation (7). When $\Delta\Phi_{DP}$ values satisfy the threshold of $0 \leq \Delta\Phi_{DP} \leq 5$ deg, pixel pairs where the absolute difference of reflectivity between the CINRAD/SA radar pixel and the X-POL radar pixel is less than 10 (i.e., $|\Delta Z(S\text{–}X)| < 10$ dB) are used for regression. On the other hand, when $\Delta\Phi_{DP}$ values satisfy the thresholds of $\Delta\Phi_{DP} > 5$ deg, all pixel pairs are involved in the regression. In addition, the $CC_{NH}$ must be more than 0.6, which is not a necessary condition for static RIC but for dynamic RIC.

$$CC_{NH} = \frac{\sum_{i=1}^{n}\left(\left(\Delta\Phi_{DP}(i) - \left(\sum_{i=1}^{n}\Delta\Phi_{DP}(i)\right) \div n\right) \times \left(\Delta Z(i) - \left(\sum_{i=1}^{n}\Delta Z(i)\right) \div n\right)\right)}{\sqrt{\sum_{i=1}^{n}\left(\Delta\Phi_{DP}(i) - \left(\sum_{i=1}^{n}\Delta\Phi_{DP}(i)\right) \div n\right)^2 \times \sum_{i=1}^{n}\left(\Delta Z(i) - \left(\sum_{i=1}^{n}\Delta Z(i)\right) \div n\right)^2}} \tag{6}$$

$$RMSE = \sqrt{\frac{\sum_{i=1}^{n}\left(\Delta Z(i) - (a \times \Delta\Phi_{DP}(i) + \Delta Z_0)\right)^2}{n-1}} \tag{7}$$

(d)　Calculations of $Z_D$ and $Z_W$:

The reflectivity pixels arrive at the X band radar site to determine whether the X band antenna cover is dry or wet. This arrival time is recorded. Then, the $\Delta Z_0$ values of a dry antenna cover and a wet antenna cover are respectively obtained by static RIC (an hour) with the arrival time as the boundary. The difference between the dry state $\Delta Z_0$ and the wet state is the attenuation caused by the water layer on the antenna cover. Moreover, $\Delta Z_0$ in a dry state is $Z_D$.

(e)　Evaluation of rain attenuation correction:

In this part, the corrected results of static RIC and dynamic RIC are compared with the ones of empirical DP relation suggested by Bringi et al., 1990 [3] to determine the best-fitted rain attenuation correction method. All volume scanned data during an entire rainfall process are divided into two groups. The first group of data is used to form a pair of correction coefficients (ratios *a* and intercepts $\Delta Z_0$) by dynamic RIC, static RIC (an hour), and static RIC (a rain event), respectively. The second group of data uses the above-obtained coefficients to correct rain attenuations, water layer attenuations, and systemic reflectivity biases. In the second group, the X-band reflectivity before (after) correcting rain attenuation and the observed S-band reflectivity, respectively obtained from the two pixels in every pairs, are used to calculate the mean absolute deviation (*MAD* in Equation (8)), standard deviation (*SD* in Equation (9)), and correlation coefficient (*CC* in Equation (10)). Then the results of *MAD*, *SD* and *CC* before and after correction are compared to show the correction effect of different correction methods. Meanwhile, the reflectivity of X band volume scanned data must firstly correct the $\Delta Z_0$ of systemic reflectivity bias and/or water layer attenuation derived through using dynamic or static RIC.

$$MAD = \frac{\sum_{i=1}^{n}\Delta Z(S-X)}{n} \tag{8}$$

$$SD = \sqrt{\frac{\sum_{i=1}^{n}\left((Z_S(i) - Z_X(i)) - \left(\frac{\sum_{i=1}^{n}Z_X(i)}{n} - \frac{\sum_{i=1}^{n}Z_S(i)}{n}\right)\right)^2}{n}} \tag{9}$$

$$CC = \frac{n\sum_{i=1}^{n} Z_X(i) \times Z_S(i) - \sum_{i=1}^{n} Z_X(i) \times \sum_{i=1}^{n} Z_S(i)}{\sqrt{\left(n\sum_{i=1}^{n}(Z_X(i))^2 - \left(\sum_{i=1}^{n} Z_X(i)\right)^2\right) \times \left(n\sum_{i=1}^{n}(Z_S(i))^2 - \left(\sum_{i=1}^{n} Z_S(i)\right)^2\right)}} \tag{10}$$

## 3. Results

### 3.1. Ratio a

Table 2 lists the values of ratio *a* found through theoretical simulations assuming a mean drop aspect ratio (i.e., minor-to-major dimension ratio) vs. size relations and measured rain temperature in the radar site and using the measurements of the dual-wavelength radar with dual-polarization capabilities introduced by Matrosov et al., 2014 [11]. From Table 2, it can be seen that the values of ratio *a* generally vary in the range from 0.1 to 0.3. As a consequence, a large correction error for rain attenuation, positively proportional to the differential phase at X band, is induced if a "false" ratio *a* in Table 2 is selected and used.

**Table 2.** X band ratios *a* from scattering simulations and experiments.

| Article | *a* (dB deg$^{-1}$) | Temperature (°C) | Mean Drop Aspect Ratio–Size Relations |
|---|---|---|---|
| Bringi et al., (1990) [3] | 0.247 | 15 | Green et al., (1975) [22] |
| Jameson et al., (1992) [7] | 0.248–0.195 | 0–40 | Pruppacher et al., (1970) [23] |
| Testud et al., (2000) [10] | 0.315 | | Keenan et al., (1997) [24] |
| Matrosov et al., (2002) [8] | 0.22 | 5 | Pruppacher et al., (1970) [23,24] |
| Park et al., (2005) [9] | 0.173–0.315 | 15 | Keenan et al., (2001) [25], Andsager et al., (1999) [26], Park et al., (2005) [9] |
| Kim et al., (2010) [27] | 0.1–0.6 | 0–30 | Anagnostou et al., (2008) [28], Matrosov et al., (2005) [29] |
| Matrosov et al., (2009) [30] | 0.23–0.28 | | Beard et al., (1987) [31] |
| Snyder et al., (2010) [32] | 0.313 | 10 | Brandes et al., (2002) [33] |
| Matrosov et al., (2014) [11] | 0.20–0.31 | | |

The data with the heavy rain event on 20 July 2016 (Beijing Time) observed by four X-POL radars and a CINRAD/SA radar are used to determine the optimal ratio *a* of linear $\Delta Z(S–X)$-$\Delta\Phi_{DP}$ relation. To take BJXFS site as an example, Figure 3a–c shows the best-linear-fit line for $\Delta Z(S–X)$-$\Delta\Phi_{DP}$ relations by dynamic (or static) RIC methods. Table 3 lists the best-linear-fit relations in Figure 3a–c and the values of fitted parameters $CC_{NH}$ and *RMSE*. It is found that the difference between dynamic RIC, static RIC (an hour), and static RIC (a rain event) is the number of volume scanned data or the number of pixel pairs. In this example, the dynamic RIC could spend the shortest time (e.g., once volume scan) in obtaining ratio *a*, and its values of $CC_{NH}$ and *RMSE* are superior to those from the static RIC. In addition, it was also found that ratio *a* is a variable parameter in this rain event.

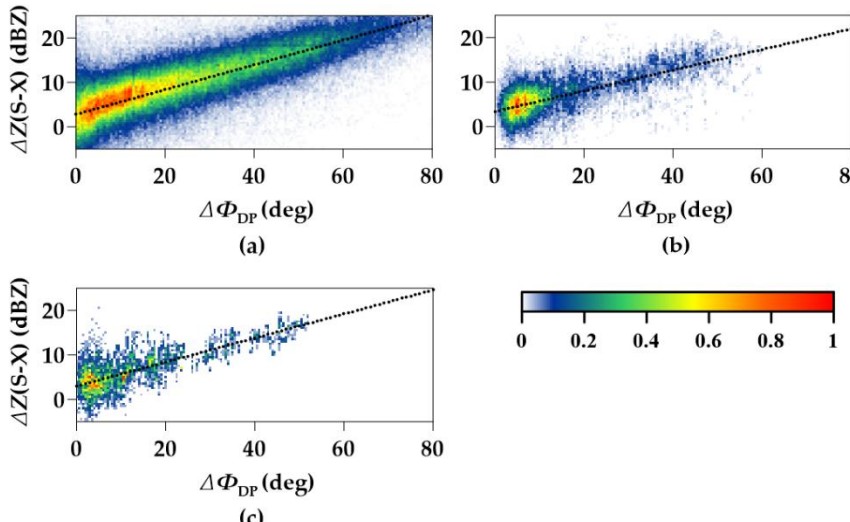

**Figure 3.** Frequency plots of observed S band–X band of reflectivity vs. X band differential phase after correcting $\Phi_{DP}(0)$: (**a**) example of static RIC (a rain event) using data on the entire rainfall event of 20 July 2016; (**b**) example of static RIC (an hour) using data from 05:48 to 06:48 Beijing time; (**c**) example of dynamic RIC using data before the 06:16 Beijing time. The black dotted line in (**a**–**c**) are the optimal regression lines.

**Table 3.** Estimated coefficients in $\Delta Z(S–X) = a \times \Delta \Phi_{DP}(X) + \Delta Z_0$ relations by static or dynamic RIC.

| Fitting Method | Fitting Relations | $CC_{NH}$ | RMSE | Data Number |
|---|---|---|---|---|
| Static RIC (a rain event) | $\Delta Z(S–X) = 0.279 \times \Delta \Phi_{DP}(X) + 2.8$ | 0.80 | 4.4 | 266 |
| Static RIC (an hour) | $\Delta Z(S–X) = 0.232 \times \Delta \Phi_{DP}(X) + 3.4$ | 0.66 | 3.4 | 8 |
| Dynamic RIC | $\Delta Z(S–X) = 0.270 \times \Delta \Phi_{DP}(X) + 3.0$ | 0.76 | 2.8 | 1 |

Figure 4a shows temporal variations of ratio *a* derived through dynamic and static RIC under the condition of $CC_{NH} > 0.60$ for four X-POL radars during this rainfall period. The values of ratio *a* derived through dynamic RIC show slight variations between adjacent time of volume scans at the same radar site. There is also difference in spatial distributions of rainfall regions observed by different X-POL radars at the same time. Despite the above disparities, after averaged in an hour, these ratios are quite consistent with the ones of static RIC (an hour). After calculating the average bias between the two kinds of average ratios, it was found that the values of the bias are 0.003 for BJXFS site, −0.001 for BJXFS site, 0.002 for BJXSY site, and −0.009 for BJXTZ site, respectively. Meanwhile, those ratios of dynamic RIC after averaged in a rainfall event also show a slight difference compared with the ones of static RIC (a rain event). Namely, the bias is −0.011 for BJXCP site, −0.023 for BJXFS site, −0.004 for BJXSY site, and −0.026 for BJXTZ site, respectively. As a consequence, the ratio *a* of the dynamic RIC not only can be calculated in real time but also is stable and appropriate. In addition, the temporal variations of fitting parameters $CC_{NH}$ and *RMSE* are also shown in Figure 4b. It can be seen that $CC_{NH}$ has a negative correlation with *RMSE*, especially when rainfall regions are away from the radar site. Compared with Figure 4a, at the same time, if there are higher values of $CC_{NH}$ or lower values of *RMSE*, ratio *a* of Equation (4) is more accurate.

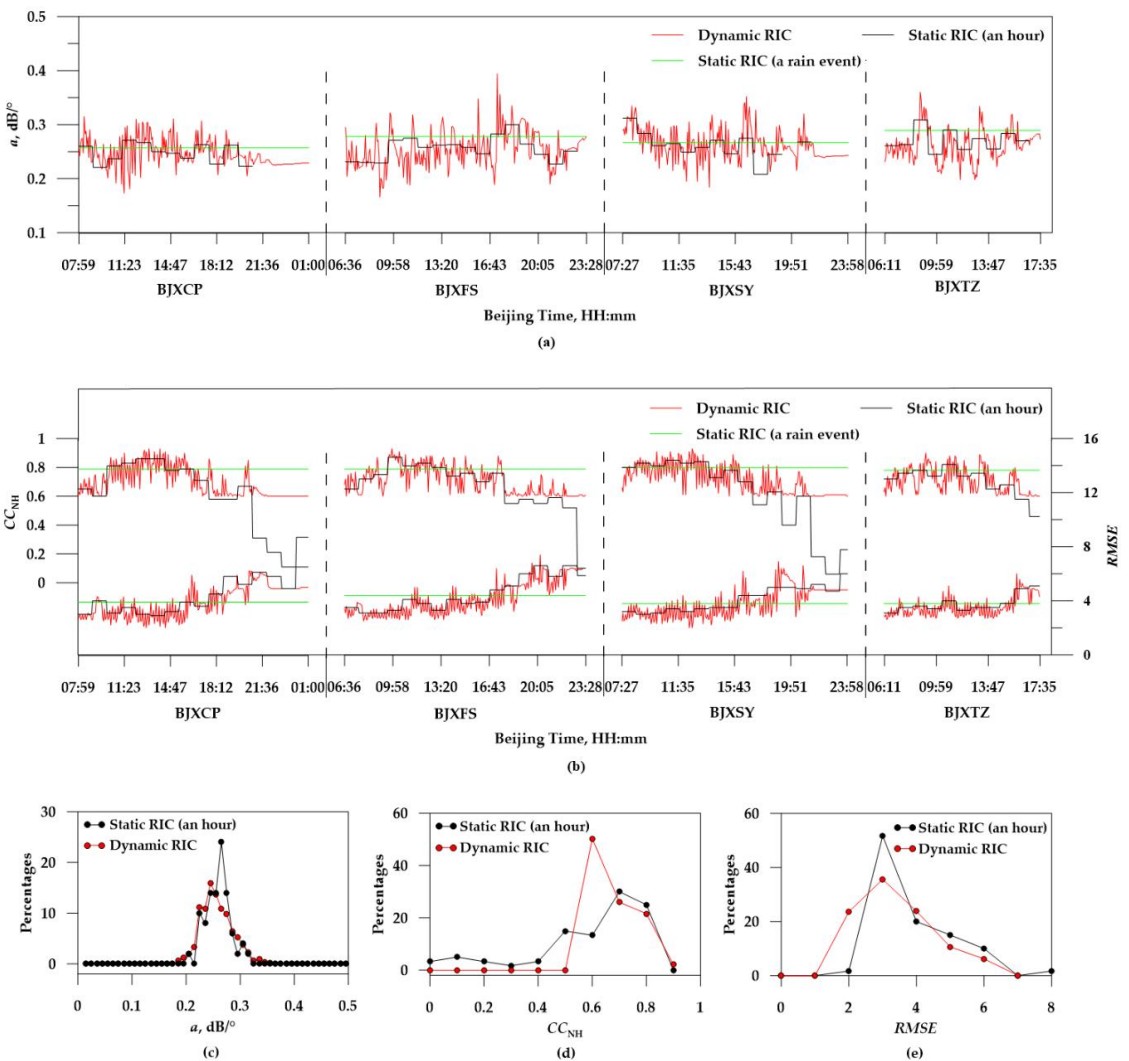

**Figure 4.** (**a**) Temporal variations of ratio *a* of BJXCP, BJXFS, BJXSY, and BJXTZ sites in the rain event on 20 July 2016 (Beijing Time). (**b**) Temporal variations of correlation coefficient ($CC_{NH}$; top) and root-mean-square error (*RMSE*; bottom). (**c**–**e**) Statistics of ratio *a* in (**a**) and parameters $CC_{NH}$ and *RMSE* in (**b**), respectively.

The statistics of these values of ratio *a*, parameters $CC_{NH}$ and *RMSE* are shown in Figure 4c–e, respectively. In Figure 4c, where the rain event data observed in Beijing on 20 July 2016 (Beijing time), are used, the obtained values of ratio *a* through using dynamic and static RIC methods vary from about 0.20 to 0.31, with a mean value of 0.26 and a standard deviation of 0.03. Meanwhile, Figure 4d,e shows the values of $CC_{NH}$ and *RMSE* from dynamic and static RICs that vary above 0.60 and a mean value of about 3, respectively, with a few exceptions. Such rainfall regions are away from the radar site so rainfall intensity near the radar site is much lower.

Similar to the trends of parameters $CC_{NH}$ and *RMSE* in Figure 4b, as shown in Figure 5, when the amount of rainfall in the regions between X-POL and CINRAD/SA radar sites maintains relatively high levels, the number of volume scanned data required to obtain real-time ratios *a* of Equation (4) is only one or two. Thereby, the above ratios *a* may have a strong correlation with current rain intensity distributions. On the other hand, it is not expected that the amount of rainfall decreases to approximately 0 value resulting in a rapid increase in the number of data. Hence, there may be a weak correlation between the real-time obtained ratio *a* and current rain intensity distributions.

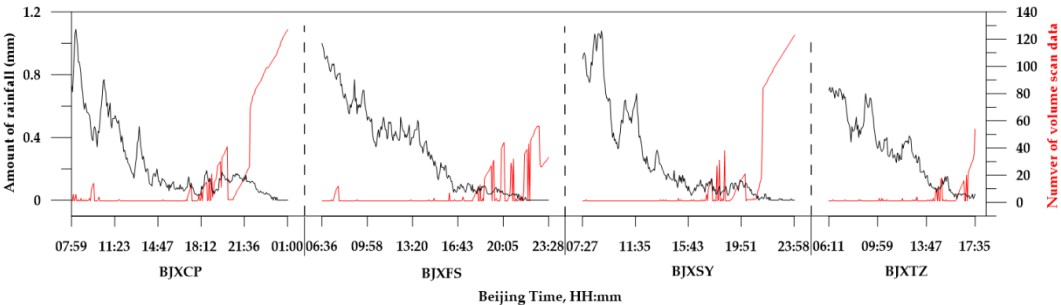

**Figure 5.** Temporal variations of the average amount of rainfall (black curve) in the regions between X-POL and CINRAD/SA radar sites and the needed number of volume scanned data (red curve) for obtaining real-time ratios *a* through using dynamic RIC. The rainfall data were collected by automatic meteorological observation stations (AMOSs) in the rain event on 20 July 2016 (Beijing Time). The conditions were that the differences between the ranges from the AMOS site to the X-POL radar site and the ranges from the AMOS site to the CINRAD/SA radar site were less than 10 km. Between the covered observation regions of BJSDX and BJXCP sites (BJXFS, BJXSY, or BJXTZ site), 166 AMOSs (158 AMOSs, 105 AMOSs, and 177 AMOSs) were obtained.

## 3.2. Reflectivity Attenuation of Water Layer on Antenna Cover

Attenuation $Z_W'$ (unit: dB) in the water layer on the X-POL radar antenna cover is related to water layer thickness (*r*, unit: m) and radar wavelength ($\lambda$, unit: m) [5]. The theoretical expressions can be written as:

$$Z_w' = 2 \times \frac{2\pi}{\lambda} \times \varepsilon'' \times r \tag{11}$$

where conductivity-related $\varepsilon''$ (unit: dimensionless) is connected to medium temperature and is calculated by Debye formula [34]. Assuming that raindrops fall vertically onto the spherical antenna cover and without splashing, *r* can be expressed as [5]:

$$r = 0.0082 \times \sqrt[3]{R \times d} \times 10^{-3} \tag{12}$$

where the *R* (unit: mm h$^{-1}$) is accumulation precipitation per hour collected by AMOSs at the X-POL radar site and *d* is the antenna cover diameter of the X-POL radar whose value is equal to 450 cm.

Note that the variations of theoretical $Z_W'$ in Equation (11) only result from the differences in *R* in Equation (12) between adjacent hours. Figure 6 shows the temporal variations in the relationship between theoretically calculated $Z_W'$ and experimentally estimated $Z_W$ in the rain event on 20 July 2016. Although the trends of $Z_W'$ and $Z_W$ are slightly different, $Z_W$ is constant most of the time and its values mainly vary in the interval of $Z_W'$, with some exceptions in BJXFS. In addition, found in calculations for theoretical values of water layer attenuation $Z_W'$ from Figure 6 that are generally less than 1 dB and smaller than the values of rain attenuation at the same time. For example, in a moderate rainfall, when the *R* of the BJXCP site is about 20 mm h$^{-1}$ in the 12th hour and $Z_W'$ derived from Equation (11) and Equation (12) is only 0.8 dB, the values of calculated rain attenuation from Equation (2) vary from 3 to 26 dB. Moreover, in a light rainfall, *R* of about 5 mm h$^{-1}$ at the BJXCP site during the 8th hour has the $Z_W'$ value of 0.4 dB. Meanwhile, the values of rain attenuation vary from 5 to 18 dB. Therefore, the values of water layer attenuation $Z_W'$ (and $Z_W$) are generally smaller than the values of rain attenuation.

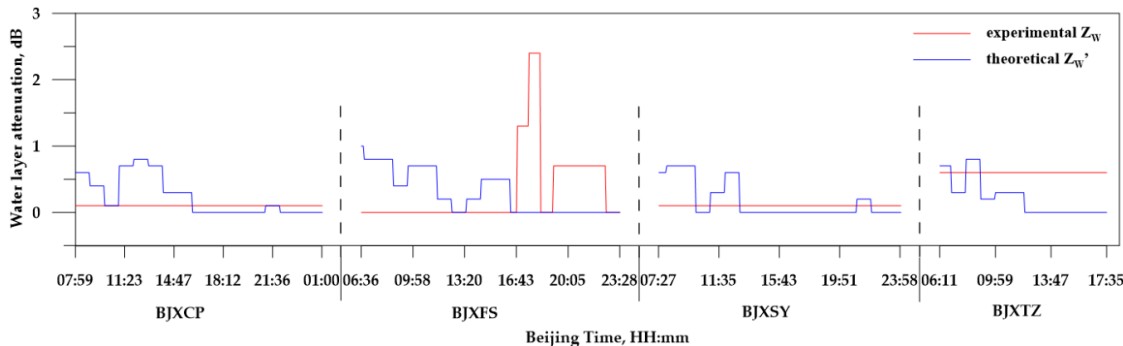

**Figure 6.** Temporal variations of experimentally estimated $Z_W$ through using static reflectivity integrated correction (RIC) (an hour) and of theoretically calculated $Z_W'$ through using rainfall rate $R$ collected at BJXCP, BJXFS, BJXSY, and BJXTZ sites in the event on 20 July 2019.

### 3.3. Systemic Reflectivity Bias between X-POL and CINRAD/SA Radars

Pixels are identified in rainfall regions between the adjacent CINRAD/SA radar and the X-POL radar in a multi-band radar network, specifically those that have the same locations in three-dimensional space. Then, the evaluation of calibration differences between two radars is carried out through making long-term comparisons between matched reflectivity pairs. A similar idea was introduced earlier by Gourley et al., 2003 [35] and has been extensively applied in the American WSR-88D radar network. Substituting the estimated $Z_W$ values into Equation (5) and setting a condition of $CC_{NH} > 0.6$, the temporal variations of systemic reflectivity bias between X-POL and CINRAD/SA radars (i.e., $-Z_D$) in this rain event is shown in Figure 7a. It reveals that $-Z_D$ values consistently fluctuate within an average range at four X-POL radar sites (BJXCP, BJXFS, BJXSY, and BJXTZ). The values are 0 dB between BJXCP and BJSDX sites, $-3$ dB between BJXFS and BJSDX sites, 2 dB between BJXSY and BJSDX sites, and 0 dB between BJXTZ and BJSDX sites, respectively. The absolute difference of <3 dB between four X-POL radars may result from the reflectivity discrepancies caused by variable refractivity gradients yielding different beam propagation paths, differing radar cross-sections dependent on the viewing angle for some hydrometeors [35], and increasing differences in beam volumes between CINRAD/SA and X-POL radars along the propagation path. In addition, the statistics for variable ranges of $-Z_D$ values from four X-POL radars are shown in Figure 7b–e. The distributions of $-Z_D$ values of BJXCP and BJXFS focus on the ranges whose absolute value is less than 1. Moreover, $-Z_D$ values vary from 0 to 3 and from $-5$ to 0 in BJXSY and BJXFS sites, respectively. Although the four X-POL radars have the same technology parameters, BJXCP and BJXTZ sites have better stable radar system performances than BJXFS and BJXSY sites.

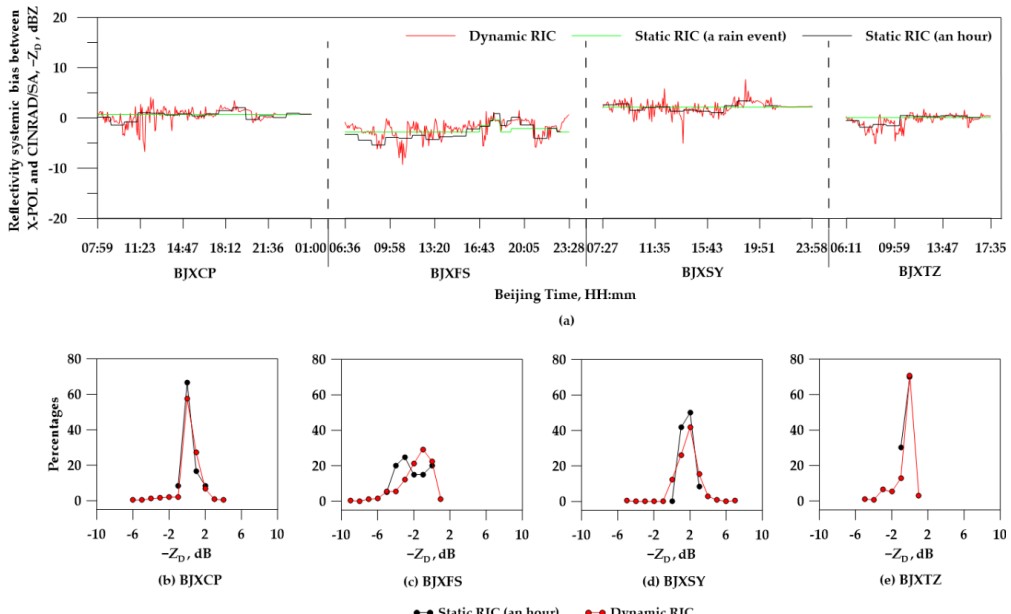

**Figure 7.** (**a**) Temporal variations of systemic reflectivity bias between X-POL and CINRAD/SA radars ($-Z_D$) in the rain event on 20 July 2016. The statistics for variable ranges of $-Z_D$ at four X-POL radar sites: (**b**) BJXCP, (**c**) BJXFS, (**d**) BJXSY, and (**e**) BJXTZ.

## 4. Evaluations of Rain Attenuation Correction

The data from the rain event that occurred on 20 July 2016, are evaluated in 1 h segments, using RIC algorithms and an empirical DP relation (empirical ratio $a$ = 0.247). The event was continually observed for 16 h by BJXCP, 16.5 h by BJXFS, 15.5 h by BJXSY, and 10.5 h by BJXTZ, which was about 59 h in total. There is an example showing that selecting 1 h data from 0548 to 0648 on 20 July 2016, at BJXFS makes corrections for visually representing the corrected effects for rain attenuation. The corrected ratios $a$ and intercepts $\Delta Z_0$ in Table 3 and others (dynamic RIC ratios $a$ and intercepts $\Delta Z_0$: 0.279, 2.9 at 0552 data; 0.229, 3.6 at 0600 data; 0.238, 3.6 at 0608 data; 0.233, 2.6 at 0624 data; 0.225, 3.3 at 0632 data; 0.217, 3.6 at 0640 data; and 0.216, 3.5 at 0648 data) are used to correct another half-hour data. The frequency plots of X band reflectivity $Z_X$ before and after rain attenuation correction vs. observed S band reflectivity $Z_S$ are shown in Figure 8a,b.

Note that in Figure 8a, the reflectivity pairs from X and S band radars are not symmetric along the diagonal and the pairs are offset to the $Z_S$ axis side. The offset phenomenon occurs at large reflectivity, because the X band attenuation is more severe than the S band one due to the shorter wavelength of the X band radar. The calculation of the value of *MAD* (*SD* and *CC*) is 3.4 (4.7 and 0.43) before the correction. However, the offset is well decreased and even eliminated after rain correction through dynamic RIC, as shown in Figure 8b. This figure shows that the reflectivity pairs are symmetric along the diagonal and they mainly focus on the diagonal. The quality of X band reflectivity is improved to a large extent. After the correction, the *MAD* (*SD*) value drops to 0.0 (3.7) and *CC* value rises to 0.68. In addition, similar to Figure 8b, the difference between the empirical relation suggested by Bringi et al., 1990 [3] and the determined best-linear-fit relations using RIC algorithms is not at image, at the values of *MAD* (*SD* and *CC*).

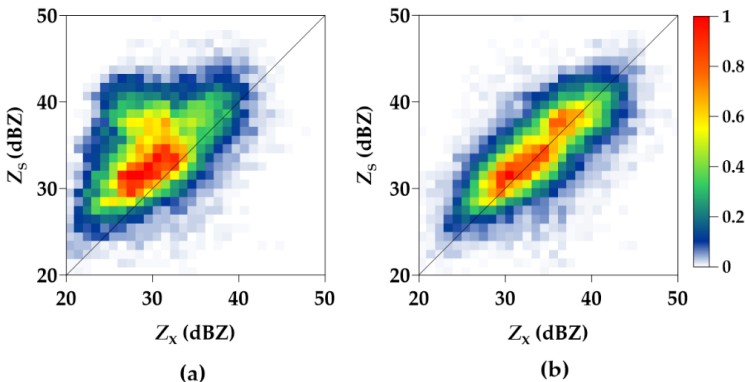

**Figure 8.** Frequency plots of X band reflectivity ($Z_X$) before and after rain attenuation correction vs. observed S band reflectivity ($Z_S$): (**a**) before the correction and (**b**) after the correction by dynamic RIC. The other frequency plots of rain attenuation correction are similar to (**b**) and are omitted here; such are static RIC (a rain event) or static RIC (an hour) and an empirical DP relation.

Therefore, the total of 59 h data are used for statistics of the variable range of *MAD, SD,* and *CC* obtained using dynamic RIC, static RIC (an hour), static RIC (a rain event), and the empirical relation. There are only 51 pairs of ratios *a* and $\Delta Z_0$ intercepts reaching the condition of $CC_{NH} > 0.6$ during 59 h in total while using static RIC (an hour). Meanwhile, the probability distributions of the above statistical results are shown in Figure 9a,c. It is obvious that the correction of rain attenuation helps a lot in the improvement of the values of *CC* and *SD,* and especially *MAD.* Before the correction of rain attenuation, Figure 9 not only shows that the values of *CC* (*SD*) widely vary from 0 to 0.8 (from 3 to 10) and that its mean value is 0.35 (5.8) but also shows that the values of *MAD* are distributed in the wide range of −1 to +13 with a mean value of 5.5. On the other hand, after the correction of rain attenuation, the variations of *CC, SD,* and *MAD* parameters are narrowed down to the accepted range, where their values decrease to the range of 0.4–0.8 (2–10 and −2 to +3), with a mean value of 0.62 (4.3 and 0.2).

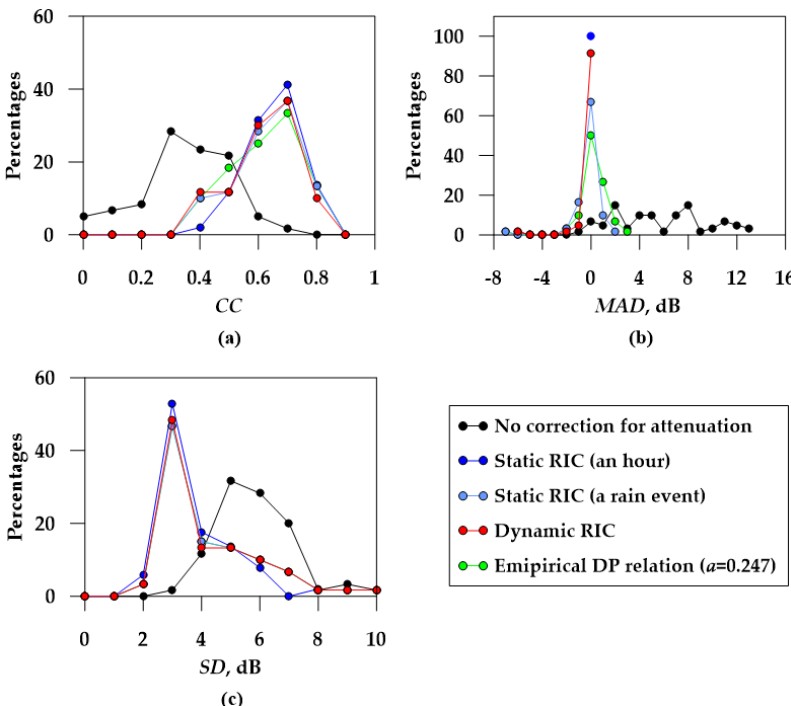

**Figure 9.** Probability distribution graphs of corrected results of five correction methods using 59 h segment data from 20 July 2016: (**a**) *CC,* (**b**) *MAD,* and (**c**) *SD.*

In addition, there are some discrepancies between the trends of corrected parameters obtained using different correction methods. Table 4 shows the comparison of corrected trends and uncorrected trends in Figure 9a,c, calculating the improved percentage values of *CC, SD,* and *MAD* of different correction methods under conditions $CC > 0.7$, $SD \leq 3$, and $MAD = 0$. The usage of ratios *a* of Equation (4) in rain attenuation correction, which are obtained in a shorter period of time, renders higher correction accuracy resulting from considering spatial–temporal changes in the microphysical characteristics of the particles in the rainfall. For example, the improved *MAD* value of dynamic RIC and static RIC (an hour) in Table 4 is about twice as large as the one of empirical DP relation with the ratio of 0.247 dB deg$^{-1}$ and is about four thirds as large as the one of static RIC (a rain event). Furthermore, the improved corresponding *CC* (*SD*) value is also large (small).

**Table 4.** Improved percentage differences between results after attenuation correction and results before attenuation correction.

| Parameter Interval | Dynamic RIC (%) | Static RIC (An Hour) (%) | Static RIC (A Rain Event) (%) | Empirical Ratio $a = 0.247$ dB deg$^{-1}$ (%) |
|:---:|:---:|:---:|:---:|:---:|
| MAD = 0 | 85.0 | 100.0 | 60.0 | 43.3 |
| CC > 0.7 | 46.7 | 52.9 | 48.3 | 44.9 |
| SD ≤ 3 | 49.9 | 56.8 | 48.3 | 48.3 |

## 5. Conclusions

Ratio $a = A_{\mathrm{H}}/K_{\mathrm{DP}}$ is usually obtained through theoretical simulations that assume certain relations between the drop aspect ratio and size, as well as averaged measured temperature [3,7,36] with some exceptions [11,37]. The determined values of *a* obtained using the methods explained above have some practical limitations in rain attenuation correction. For example, it does not conform to the characteristics of temporal–spatial variability of actual raindrop distributions but only highly approaches the average value in a rain event. A best-linear-fit modified attenuation correction algorithm based on differential phase $\varPhi_{\mathrm{DP}}$ and attenuation is given and applied for a moderate long-time rain event, collected by four X-POL radars and a CINRAR/SA radar in Beijing.

The proposed correction algorithm is applied in attenuation correction for pure rain, and it renders a reliable *a* ratio of $A_{\mathrm{H}}/K_{\mathrm{DP}}$ for every X band radar volume scanned data in real time. The values of ratio *a* are consistent with those previously reported in literature, with a mean value of 0.26 varying from 0.20 to 0.31. Moreover, the proposed correction algorithm, under the premise that reflectivity is unattenuated in S band radar detection, also considers the systemic reflectivity bias between two kinds of wavelength radars and additional attenuations caused by the water layer over the antenna cover. On the one hand, the statistical analysis of water layer attenuation of four X band radars shows that the temporal variability of experimentally fitted attenuation values is in substantial agreement with the theoretically calculated ones. The simultaneous comparison of the water layer attenuation and rain attenuation also reveals that the former is an order of magnitude smaller than the latter, with the former being less than 1 dB. On the other hand, the average values of systemic reflectivity bias between the X band radar sites (BJXCP, BJXFS, BJXSY, and BJXTZ) and S band radar site (BJSDX) are 0, −3, 2, and 0 dB, respectively. Finally, the proposed correction algorithm for rain attenuation has higher correction accuracy, which was testified in 59 h periods through comparisons with the usage of empirical ratio *a* at X band suggested by Bringi et al., 1990 [3].

Although the correction algorithm proposed in this study represents a new approach for reliably obtaining the ratio *a* of $A_{\mathrm{H}}/K_{\mathrm{DP}}$ in real time and its good performance in rain attenuation correction has been verified, there are still some restrictions. For the first time, when obvious attenuation exists in the cross-observation rainfall regions of X and S band radars, the algorithm has good correction performance. Then, the differences in beam volumes between CINRAD/SA and X-POL radars along the propagation path rapidly increase due to the range resolution disparity of CINRAD/SA radar of

1 km and the X-POL radar of 75 m. That may result in a collection of pixel pairs with a large difference in reflectivity. Finally, considering the temporal–spatial variability of RSD, a CINRAD/SA radar should be combined with a multiple X band radar, which may effectively correct X band reflectivity in the whole observation area, including the effect of rain attenuation, water layer attenuation, and systemic reflectivity bias.

**Author Contributions:** Conceptualization and methodology, C.W. (Chong Wu) and L.L.; formal analysis, investigation, and writing—original draft preparation C.W. (Chao Wang); writing—review and editing, C.W. (Chao Wang), C.W. (Chong Wu), L.L., X.L., and C.C.; supervision, C.W. (Chong Wu) and L.L.; funding acquisition, C.W. (Chong Wu) and L.L. All authors have read and agreed to the published version of the manuscript.

**Funding:** This work was partially supported by the National Key Research and Development Plan of China (Grant No. 2018YFC1507401 and Grant No. 2018YFC1506101).

**Conflicts of Interest:** The authors declare no conflict of interest.

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
