# Peer review of "Integrated Correction Algorithm for X Band Dual-Polarization Radar Reflectivity Based on CINRAD/SA Radar"

_atmosphere, doi:10.3390/atmos11010119_

Round 1

Reviewer 1 Report

This paper provides a correction algorithm for the X band dual-polarization radar reflectivity. It is a good contribution for the whole community. I recommend it publication in the present form.

There are several minor comments about grammar

Line 39-41  This sentence is too long, and with obvious grammar error, please decompose it into two shorter sentences

Line 117  remove ‘respectively’

Line 174  change ‘by using’ into ‘by’

Line 215 change ‘scan’ into ‘scanned’

Line 220-223 this sentence is too long, please decompose it into two shorter sentences

Line 279-281  what is ‘high levels’, could you give an explanation??

Line 314  change ‘differential’ into ‘different’

Author Response

Thank you very much for your comments, I have uploaded my response in the attachment, please see it.

Reviewer 2 Report

The manuscript by Wang and co-authors describes a new scheme to correct X band radar reflectivity data using an S band radar in the Beijing area. The idea is to use the S band radar data in the integrated correction algorithm which works on the basis of a linear relationship between specific attenuation and specific differential phase. The work is properly set into the context of previous research, is generally well written, explains in adequate detail the new scheme and the observations that have been used in the application of the scheme to real data. The statistical analysis of the results is convincing. I have a small observation to make regarding the definition of rain volumina measured by both S and X band radar (see section 2.2 and Figure 2). The viewing angle difference between the two radars is defined to be less that 0.01 degrees. This seems to be very stringent. Could the authors perhaps double-check this parameter? I consider the work to be of sufficient originality to be published.

Author Response

(The authors gave the same response as above.)
